# Determinants of COVID-19 Knowledge, Perception and Attitudes in Indonesia: A Cross-Sectional Survey

**DOI:** 10.3390/ijerph20043713

**Published:** 2023-02-20

**Authors:** Al Asyary, Meita Veruswati, Rony Darmawansyah Alnur, La Ode Ahmad Saktiansyah, La Ode Hasnuddin S. Sagala, Syawal Kamiluddin Saptaputra, Eka Oktavia, Maria Holly Herawati, Ririn Arminsih Wulandari, Hanns Moshammer

**Affiliations:** 1Department of Environmental Health, Faculty of Public Health, Universitas Indonesia, Depok City 16424, Indonesia; 2Study Program of Public Health, Faculty of Health Sciences, University of Muhammadiyah Prof. Dr. HAMKA (UHAMKA), Jakarta 12130, Indonesia; 3PhD Program of Business and Management, Postgraduate School, Management and Science University (MSU), Shah Alam 40100, Selangor Darul Ehsan, Malaysia; 4Study Program of Public Health, Faculty of Public Health, Halu Oleo University, Kendari 93232, Indonesia; 5Department of Information System, Faculty of Information Technology, University of SembilanBelas November, Kendari 93561, Indonesia; 6Alifa Pringsewu Midwifery Academy, Kota Bandar Lampung 35373, Indonesia; 7Indonesian National Research and Innovation Agency (BRIN), Jakarta 40173, Indonesia; 8Center for Public Health, Department of Environmental Health, Medical University of Vienna, 1090 Vienna, Austria; 9Department of Hygiene, Medical University of Karakalpakstan, Nukus 230100, Uzbekistan

**Keywords:** COVID-19, knowledge, perception, health literacy, Indonesia

## Abstract

Many countries, including Indonesia, were gravely affected by the COVID-19 pandemic. While younger people were rarely severely affected by an infection, they still served as important spreaders of the disease. Therefore, the knowledge, perception and attitudes regarding COVID-19 of a mostly younger population was assessed in this study using a quantitative survey and semi-structured questionnaire. Out of 15 questions on COVID-19, males answered fewer questions (−1.26) correctly. Persons reporting more diseases in the last year (+0.49 per disease) that lived in a central area of Indonesia, and that had a better socio-economic status defined through household condition scores, had better knowledge of the symptoms, causes of and measures against COVID-19. Better knowledge independently predicted more responsible attitudes and stated behavior. Knowledge and understanding should be enhanced through information campaigns targeted specifically towards men, persons with poor socio-economic backgrounds and those living in the periphery of the state.

## 1. Introduction

Similar to many other countries around the world, Indonesia was also severely affected by the COVID-19 pandemic [1,2,3]. Since therapeutic options were limited [4], the main focus was on preventive strategies. In the earlier stages of the pandemic, these measures focused on social distancing and protection by the use of face masks, especially [5], and later, as soon as vaccines became available, vaccination campaigns were launched. Especially younger and healthy people are often not severely affected by COVID-19. In this population segment, infections typically occur with only minor symptoms [6]. This poses a challenge for diagnosis, and, since also infected people without symptoms can infect other people, undiagnosed infections increase the risk of the transmission and spread of the disease [6]. Indonesia is a developing country with restricted resources for public health. Therefore, the financial burden of prevention measures has to often be conducted the individual citizen: people had to pay for their own COVID-19-infection testing [1]. The situation was quite different in European welfare states, where the government offered regular testing facilities for free (own observation). If tested positive, self-isolation usually caused economic losses to the individual that were not reimbursed by the government. Therefore, knowledge of and attitudes toward COVID-19, especially of younger and healthy people, was key in all national prevention strategies. We aim to study the related knowledge and attitudes in a mostly younger population segment of Indonesian citizens and we aim to understand how knowledge and attitudes are affected by individual traits. 

We hypothesize that better education and higher socio-economic status would lead to better knowledge about COVID-19. In addition, we expect that persons more knowledgeable about the causes and symptoms of COVID-19 and with a greater understanding of the necessary measures against the disease would also report to be more supportive of these measures.

## 2. Method

### 2.1. Study Location and Design

The cross-sectional survey was conducted in four regencies, namely, the Depok district, Tegal city and district, and Kendari district. These regencies comprise a mixture of urban and rural areas. While in the first half of 2020, the hotspots of the COVID-19 pandemic in Indonesia were the main ports and industrial hubs of Jakarta and Surabaya, later in that year, the highest incidences were observed in these four regencies covered by the survey. An online questionnaire was developed and distributed early in 2021 seeking information about personal characteristics, such as age, gender and socio-economic characteristics; knowledge and attitudes regarding COVID-19; and general health status, including health behavior and recent diseases. The questionnaire was advertised with the help of local schools.

The online questionnaire was advertised through the school official (teacher, education office in each region and headmaster) and via social media, such as WhatsApp and Telegram. The type of school varied from elementary to college, which might have introduced some bias regarding the age distribution and education status of the respondents. The questionnaire was mainly answered by students (inclusion criterion: 18 years of age minimum), teachers and parents or caregivers of students.

### 2.2. Parts and Structure of the Questionnaire

An online questionnaire tool was developed and advertised in schools of the four regencies with the highest COVID-19 incidence in Indonesia at that time. Consequently, participants mostly consisted of school staff, students and their parents. This survey was part of a series that were supported and launched by the World Class Professor (WCP) 2022 seminar series (as stated in the funding information of this paper) and advertised over the Internet.

The survey was initially designed to inform the public health authorities about which items to stress in their campaigns as well as the most effective target population, but was also later adopted to obtain scientific insights into the recognition of COVID-19 within a segment of the Indonesian population.

Individual knowledge of COVID-19 was quantified through a set of 15 questions (Appendix A). Each question allowed 5 categories of answers (fully agree/somewhat agree/do not know/somewhat disagree/fully disagree). If a statement made in the questionnaire was correct, the first two answer categories were deemed correct, and if the statement was incorrect, the last two categories. Thus, the answer to every statement was rated either correct or incorrect, and the number of correct answers (0–15) was used to measure knowledge. Thus, a higher knowledge score indicated that more statements were answered correctly.

Attitudes were assessed using 5 questions (Appendix B). In short, the willingness to undergo antigen testing, PCR testing, to self-isolate, to get vaccinated and trust in the effectiveness of vaccines, at present, were determined. In line with the knowledge part of the questionnaire, 5 categories of answers were also offered.

Socio-economic status was evaluated through different indicators. First, the respondents were asked about the number of household members and size of housing in square meters, allowing us to calculate residential population density. Second, housing conditions based on the availability of a kitchen, bedroom, living room and toilet, basic facts about these rooms and source of drinking water were assessed. Appendix C lists these items together with the scores they received. The different items were weighted according to the standard weights used in Indonesian socio-economic research [7]. The conditions regarding the bedroom, kitchen and family room were given a weight of 31; the source of drinking water, type of toilet used and place of final feces disposal a weight of 25; and the existence of a private or shared toilet a weight of 44. After applying the weights, the score that could be achieved ranged between 0 and 100. Third, the questionnaire asked for number and type of household assets (Appendix C). For example, a car received a score of 32 points and a TV set 2 points. All possible points together would provide a maximal total score of 100. Thus, both housing conditions and household assets were rated according to the Indonesian 2013 baseline health standards [8] to present a score between 0 and 100 each. The scores for each amenity are also presented in Appendix C.

For the estimation of individual health status, current smoking behavior (yes/no) and the presence or absence of a list of symptoms and diseases (respiratory infection, asthma, fever, weight loss, other diseases diagnosed by a doctor) in the last year were asked.

### 2.3. Data Analysis

Descriptive statistics are displayed either presenting the mean and standard deviation (Std. dev.) or numbers and percentages.

To examine the factors that predict knowledge, we performed linear regression analyses with knowledge as the dependent variable. Because the small sample size prevented the inclusion of all independent variables at once in a multiple regression, we decided on a two-step process: at first, each factor was tested individually. In the following step, all factors that at least displayed a tendency of influencing knowledge (*p* < 0.2) were included in a multiple linear regression. Then, factors were excluded stepwise starting with the factor with the highest *p*-value, until only factors with a *p*-value > 0.1 remained or until the removal of a factor caused the coefficient of the most influential remaining factor to change by more than 10%. A two-sided *p*-value less than 0.05 was considered significant. The results are presented in a tabular format reporting coefficients and the 95% confidence interval (CI).

To examine predictors of attitudes, we performed multinomial logistic regression using “do not know” as the base outcome. The results are presented graphically depicting the relative risk ratio and its 95% confidence interval. All analyses were performed with STATA vers. 17 [9].

## 3. Results

### 3.1. Description of the Sample

In total, 462 respondents (351 women and 111 men) answered the questionnaire. Their age ranged from 12 to 60.7 years, with age data missing for 14 respondents. Descriptive statistics of the respondents are provided in Table 1.

### 3.2. Linear Regression on Personal Factors Influencing COVID-19 Knowledge

Only a few characteristics influenced the number of correct answers significantly. Several indicators of socio-economic status (SES) were either significant or approached significance (housing condition score, *p* = 0.034; household assets, 0.083). Since these indicators were not highly correlated with each other (r = 0.191), both were initially included in the multiple regression. In the multiple regression, household assets did not remain nearly significant and contributed very little to the point estimate. Therefore, they were removed from the final analysis.

Since educational achievements are affected by age, we controlled for age in the regression analysis of the impact of education. In this additional model (not shown in Table 2), age was nearly significant (coefficient = −0.066, *p* = 0.074). However, in the multiple regression, neither age nor education affected COVID-19-related knowledge.

### 3.3. Multinomial Logistic Regression Examining Determinants of Attitudes

Knowledge influenced the attitudes towards COVID-19. The greater the knowledge about COVID-19 of the participants, the more likely they were willing to undergo antigen or PCR testing, to self-isolate in case of a positive test result and to be vaccinated when the vaccine became available. Both trust and mistrust of the effectiveness of the available vaccine that was provided by the Indonesian government was more pronounced among better-informed people (Figure 1). Figure 1a shows the willingness to undergo antigen testing. The relative risk ratios for the responses “fully agree”/“agree”/“disagree”/“fully disagree” (compared to “do not know”) were 1.39; 95% confidence interval 1.27–1.54; 1.24, 1.13–1.36; 0.89, 0.80–0.93; and 0.80, 0.69–0.93. Figure 1b reports the results for willingness to undergo PCR testing: 1.45, 1.31–1.60; 1.29, 1.17–1.43; 0.95, 0.85–1.05; 0.85, 0.75–0.97. Figure 1c depicts the results for willingness to be vaccinated: 1.38, 1.25–1.52; 1.16, 1.05–1.27; 0.85, 0.76–0.95; 0.89, 0.80–0.98. Finally, Figure 1d depicts the results for lack of trust in government-provided vaccines: 1.02, 0.90–1.52; 1.16, 1.03–1.32; 1.15, 1.06–1.24; 1.17, 1.06–1.29. The results for willingness to self-isolate were very similar to those for willingness to undergo antigen or PCR testing and are not shown as a figure: 1.53, 1.37–1.70; 1.35, 1.21–1.51; 0.97, 0.85–1.11; 0.74, 0.61–0.90.

In addition to the knowledge of COVID-19, individual characteristics had very little direct influence on individual attitudes. Males more likely abstained from antigen testing; however, when compared to the answer “do not know” (thereafter the “neutral answer”), only the strongest “fully disagree” was significantly different (*p* = 0.018) and 4.4 times more likely among males. Agreement was less likely, but this difference did not attain significance. Higher education (after controlling for age) reduced the likelihood of a neutral answer. “Strongly agree” was about 67% more likely per educational achievement (*p* = 0.02), “agree” 2.1-fold more likely (*p* = 0.004) and “disagree” 2.8-fold more likely (*p* = 0.042). In the same model, older age increased the likelihood of a neutral answer. However, only “agree” was significantly (*p* = 0.001) less likely with increasing age. Per year, the likelihood was reduced by about 8 percent.

Both housing conditions and score of household assets increased the likelihood of “strongly agree”: housing conditions (*p* = 0.006) by about 4% per score-point; household assets (*p* = 0.018) by about 3% per point. The other answer categories neither showed a clear trend nor a significant difference from the neutral response.

Estimates for the willingness to take a PCR test were generally similar, but weaker and only rarely significant. Regarding a willingness to self-isolate, only the influence of the number of household goods was observed with a significant (*p* = 0.046) increase in the “fully agree” response of about 3% per scoring point.

Willingness to be vaccinated was less likely in men. “Strongly disagree” was 4 times more likely (*p* = 0.003) in this group. On the other hand, more household assets increased the willingness: “strongly agree” was selected about 3.7% more often per scoring point.

Mistrust of the effectiveness of the vaccine provided by the government was lowest in persons with the highest number of household goods (“strongly disagree” was 3% more likely per scoring point, *p* = 0.026) and with better housing conditions (“strongly disagree” was 4.5% more likely per scoring point, *p* = 0.012). Although not reaching significance, smokers tended to display stronger distrust when the answers from “fully agree” as 1 to “fully disagree” as 5 were considered continuous; *p* for trend was 0.005.

## 4. Discussion

This study aimed to examine (a) personal predictors of good knowledge about COVID-19 causes and symptoms, and (b) how these predictors and knowledge would affect the participants’ attitudes and behaviors in case of signs of infection.

Surprisingly, knowledge about COVID-19 was not strongly associated with educational achievements. Instead, the number of recent diseases, socioeconomic status (measured through housing conditions), area of living and female gender predicted greater knowledge scores. Attitudes towards preventive measures were not strongly and consistently affected by these individual characteristics. Therefore, for example, younger age, higher education status and, to some degree, higher socio-economic status reduced the likelihood of neutral (“do not know”) answers, but had no clear effect on either affirmative or negating positions. As opposed to these single characteristics, knowledge of COVID-19 showed a clear effect on attitudes. People who were more knowledgeable about COVID-19 causes and symptoms tended to support preventive measures more strongly than those who answered less statements correctly. Only with trust in the vaccine provided by the government, greater knowledge rather led to more pronounced statements (both agree and do not agree) and not to a clear trend. This ambivalence towards the vaccine among those that were better educated might not be without some reason. On the one hand, at that time only the Chinese vaccine was available in Indonesia. This vaccine, CoronaVac©, was marketed very rapidly while the results of large phase III trials were still missing or not yet published. The first large phase III trial only showed an overall efficacy of about 50% [10]. On the other hand, parenteral vaccines against SARS-CoV-2 were generally able to booster systemic immune response, but not a sufficient IgA response in mucous membranes. While IgA is detectable in serum after vaccination [11], it is less persistent [12] and usually does not reach the respiratory epithelium in a sufficient quantity [13]. Therefore, efficacy in preventing infections was generally low, while efficacy in preventing severe disease could still be high [9]. Maybe the question was not clear enough and so some respondents answered with infection in mind and others thinking of severe disease.

Therefore, the most important finding of this study would be that better knowledge is important for responsible health behavior that is both beneficial to oneself and the community. This special knowledge about health issues is commonly discussed under the term “health literacy” [14,15,16,17,18,19,20,21,22]. From these examples, it is well understood that improved knowledge of health and health-related behaviors and lifestyles are an important driver of health outcomes. The current study supports these findings and strengthens the evidence that health literacy is also important for responsible behavior needed for protecting the community.

Of course, it must be acknowledged as a limitation of the study that any questionnaire survey only provides information on stated behavior, but cannot measure true behavior itself. This would call for a completely different approach. Nevertheless, even if we accounted for a tendency of reporting more acceptable behaviors, the associations observed would still hold as long as the reporting bias was non-differential. Nonetheless, respondents with greater knowledge would likely also know better what would be desirable. We could examine the relation between knowledge and attitude, but not between attitude and actual deeds.

Access to health information has an important role in determining health literacy [19,20,21]. However, similar to numerous other countries, access to health information in Indonesia is not equally available for the population throughout the country. The level of access is mainly determined by geographical location and the degree of urbanization, e.g., availability of Internet and electricity [21]. According to the previous research, the Internet is one of the main sources of health information [20,21,23]. Due to the very special socio-geographical situation in Indonesia, the countywide supply of access to the Internet is still under development.

At least 130 million Indonesians actively use social media. As one report [24] estimated, among the total population of Indonesia, which reaches 265.4 million people to date, about one half or 132.7 million people are Internet users in Indonesia, 130 million of whom are active users of social media with penetration values reaching 49% [24,25]. Based on the same research, about 90% of teens use the Internet regularly and 70 percent of them have at least one profile on social media.

It has been shown that housing status has a significant role in influencing health literacy [26]. It is very likely that the housing status may serve here as an indicator of the individual socioeconomic situation, which in turn is highly related to access to education, media and medical advice.

In many developed and developing countries worldwide, knowledge of and trust in public health-related scientific results have been shown to greatly affect the effectiveness of introduced mitigation measures. Education level is therefore one of the important determinants in controlling pandemics, such as COVID-19 [5,17,18]. Educational programs that increase individual health literacy can also effectively reduce public doubts of vaccines and encourage acceptance. Thus, the promotion strategy must be adapted for different populations, taking individual factors into account [27].

This study had clear limitations that were, to a large extent, highly related to the same socio-economic and access-related problems linked to COVID-19 knowledge and acceptance in Indonesia. Our survey covered a very special segment of the Indonesian population, characterized by their relation to school authorities, including parents, pupils and teachers. Our study also included significantly more women than men. Therefore, this study cannot very precisely assess health knowledge and acceptance of mitigation among men who constitute a large part of the total population. Nevertheless, the finding of poorer COVID-19-related knowledge among men in our study is still a major concern.

## 5. Conclusions

Health literacy and knowledge of disease risks and mechanisms among lay people is essential to keep them healthy. However, health literacy does not only serve their personal gain, as it is also necessary for leading a responsible life beneficial to the health of their peers. Knowledge and understanding should be enhanced through targeted information campaigns. Men, persons with poor socio-economic backgrounds and those living on the periphery of the state should be targeted specifically.

## Figures and Tables

**Figure 1 ijerph-20-03713-f001:**
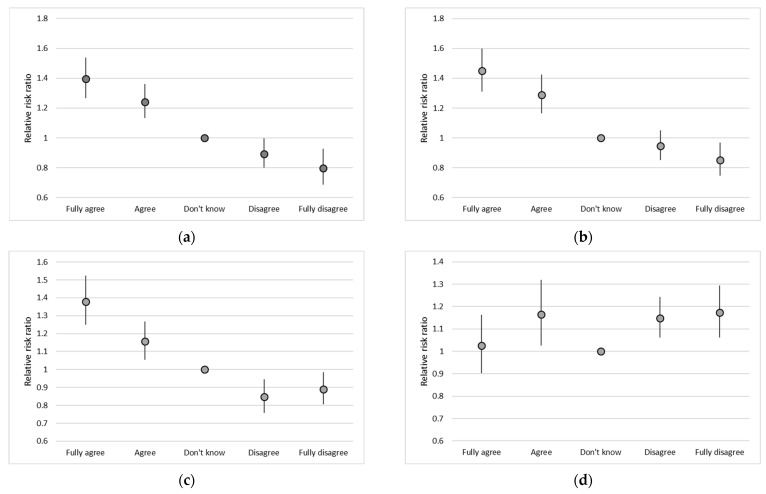
Probability of agreeing or disagreeing strongly or somewhat as compared to an undecided position (relative risk ratio) as a function of knowledge about COVID-19 (score 0–15). Point estimate and 95% confidence interval. Answer to the following questions (Appendix C): (**a**) willingness to undergo antigen testing; (**b**) willingness to undergo PCR testing; (**c**) willingness to get vaccinated; (**d**) trust in the vaccine provided by the government.

**Table 1 ijerph-20-03713-t001:** Description of study population.

Variable	All (n = 462)		Male (n = 111)		Female (n = 351)	
	Mean	Std. Dev.	Mean	Std. Dev.	Mean	Std. Dev.
Age (years)	26.04	+/−8.76	26.66	+/−8.61	25.85	+/−12.04
Socio-economic indicators						
Housing condition score	78.92	+/−8.49	80.07	+/−8.03	78.55	+/−8.61
Household assets	22.72	+/−9.76	21.35	+/−10.14	23.15	+/−9.62
Housing density (m^2^/person)	43.81	+/−239.43	37.94	+/−91.8	45.67	+/−269.9
Number of diseases	0.55	+/−0.72	0.49	+/−0.71	0.56	+/−0.72
Knowledge (0–15)	10.58	+/−3.27	9.65	+/−3.92	10.87	+/−2.98
Attitude towards AG test (1–5) *	1.87	+/−1.04	2.14	+/−1.28	1.78	+/−0.95
Attitude towards PCR test (1–5) *	1.86	+/−1.10	2.02	+/−1.23	1.81	+/−1.05
Attitudes reg. self-isolation (1–5) *	1.66	+/−1.02	1.89	+/−1.22	1.59	+/−0.94
Attitudes reg. vaccination (1–5) *	2	+/−1.20	2.13	+/−1.37	1.96	+/−1.14
Vaccine mistrust efficiency (1–5) *	3.40	+/−1.01	3.26	+/−1.13	3.44	+/−0.97
Education status	Number	Percent	Number	Percent	Number	Percent
Elementary school	18	3.90	5	4.50	13	3.70
Junior high school	6	1.30	3	2.70	3	0.85
Senior high school	225	48.70	49	44.14	176	50.14
College	213	46.10	54	48.65	159	45.30
Employment status						
Student	228	49.35	48	43.24		180, 51.28
Employed	188	40.69	58	52.25		130, 37.04
Unemployed	46	9.96	5	4.5	41	11.68
Administrative status						
Rural	108	23.43	32	29.09	76	21.65
Urban	353,	76.57	78	70.91	275	78.35
Geographical status						
Peripheral	39	8.46	16	4.65	23	19.66
Central	422	91.54	328	95.35	94	80.66
Smoker						
Yes	45	9.74	26	23.42	19	5.41
No	417	90.26	85	76.58	332	94.59

* 1: fully agree; 5: fully disagree.

**Table 2 ijerph-20-03713-t002:** Factors influencing COVID-19 knowledge (score: 0–15).

Variable	Univariate Regression		Multiple Regression	
	Coefficient	95% CI	Coefficient	95% CI
Gender				
Man	−1.22 *	−1.91; −0.53	−1.26	−1.94; −0.57
Age	−0.02	−0.06; 0.01		
Education status				
Junior high school	2.89	−0.29; 6.06		
Senior high school	0.77	−0.89; 2.41		
College	1.79	−0.06; 3.64		
Employment status				
Employed	−0.24	−0.87; 0.40		
Unemployed	−0.63	−1.67; 0.41		
Housing condition score	0.04 *	0.003; 0.073	0.046	0.011; 0.08
Household assets	0.027	−0.004; 0.058	-	-
Housing density (m^2^/person)	0.0008	−0.0005; 0.002	-	-
Administrative status				
Urban	−0.02	−0.73; 0.69	-	-
Geographical status				
Central	1.19 *	0.12; 2.26	-	-
Number of diseases	0.48 *	0.068; 0.9	0.49	0.079; 0.90
Smoker	−0.15	−1.16; 0.86	-	-

* *p*-value < 0.05.

## Data Availability

Raw data are available from the authors upon request.

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
