# Peer review of "Determinants of COVID-19 Knowledge, Perception and Attitudes in Indonesia: A Cross-Sectional Survey"

_ijerph, 2023, doi:10.3390/ijerph20043713_

Round 1
Reviewer 1 Report
Attached are my comments.
Some are related to wording and grammar.
Some comments - need to be addressed.
Please consider my remarks on lines 32, 32, 39.
The article is interesting as it sheds light on the Indonesian population's reaction to Covid, while considering important variables such as age, gender, socio-economic status, etc.

Author Response
>> Response to Reviewer 1:
Dear Reviewer 1, thank you for your positive evaluation and suggestions for our manuscript. We have now revised it well according to your comments in your attached document, not only regarding lines 32 and 39, but also most of your other comments.
Reviewer 2 Report
To the authors
Thank you for submitting this manuscript. It describes knowledge, perceptions, and attitudes to COVID-19 among a younger population in Indonesia. Across the world, we are eager to learn more about how COVID-19 was managed and gain knowledge about to manage similar situations in the future. Therefore, this study is relevant.
Below, I have provided som comments and suggestion to areas which need to be addressed.
Abstract
The abstract is short and could benefit from a concluding statement.
Introduction
Overall, the introduction is very short and needs to be extended. Actually, almost 1/3 is description of method and should be moved to the section Method (p.2, line 52-62).
The text needs to be revised and I suggest language revision by native English speaker.
p. 1, line 39: face protection; do you mean personal protective equipment or only face masks or shields? Please clarify
p. 1, line 40-41: the phrase ‘…are often not affected severely…’. Do you mean seldom?
Method
Please clarify whether the four regencies mentioned as study locations are urban or rural areas.
Overall, this section needs to be restructured, so that information about the specific questionnaire is presented in one place which should be clearly marked with a sub-heading. As it stands, information about the questionnaire is found in the introduction (se the above comment) and in the section on study location and design.
P. 2, line 76 refers to ‘in line with the attitude part of the questionnaire…’ which seems to imply, that the reader already knows about the different parts of the questionnaire. However, this information has not been provided. Please be aware of structure and clarity which is needed to follow the study transparency.
Related to most of the questionnaire information about the scores needs to be explained. Otherwise, the scores provided in Table 1 are quite difficult to follow. This includes the Indonesian baseline health standards score, please provide additional information about what low and high scores would indicate.
P. 2, line 85… please rephrase. The questionnaire is not ‘asking questions.
Analysis
Please clarify what is meant by ‘factors that at least displayed a tendency of influencing knowledge’. Please clarify the choice to include factors influencing knowledge (p= 0,2) and which p-value levels were considered to indicate a statistically significant result?
Results
As it stands, the results are structured under headings related to the statistical analysis performed. It would be more informative to present the results under headings that indicate what was actually investigated (e.g. Factors influencing COVID-19 knowledge).
The first part of the results seems to be focused more on describing the statistical analyses performed rather than focusing on what was actually found. Perhaps because the results were not statistically significant?
It would be more relevant to present actual findings and whether these were regarded as significant or not, and what this would mean in terms of clinical relevance – does it mean anything in real life?
The content of the section ‘3.3 Multinomial logistic regression’ provides much more relevant information and is much easier to follow.
Discussion
This section could benefit from language revision. Please be mindful of the language used e.g. phrasing such as ‘better knowledgeable people’.
The discussion refers to one ‘most important finding of the study’, which is ‘that better knowledge is important for responsible health behavior that is both beneficial for oneself and for the community. However, I believe it is very important to remember, that answering a questionnaire at best indicates intensions to act accordingly to what is answered. It provides no data on what the participants actually did or would do in real life. Hence you should be careful to draw any conclusions related to actions and only refer to this finding as ‘participants intentions to act’. Hence, I do not think the discussion related to health literacy is relevant. I suggest focusing discussion on issues that can help us understand why participants provided the answers they did – what might have influenced their answers – and what can actually be drawn from this study?
Conclusion
This needs to be revised based on a revision of the discussion.
Author Response
Reviewer 2
To the authors
Thank you for submitting this manuscript. It describes knowledge, perceptions, and attitudes to COVID-19 among a younger population in Indonesia. Across the world, we are eager to learn more about how COVID-19 was managed and gain knowledge about to manage similar situations in the future. Therefore, this study is relevant.
Below, I have provided som comments and suggestion to areas which need to be addressed.
>> Dear Reviewer 2, thank you for your encouragement to our manuscript that may possible to elevate our understanding about determinant of COVID-19 further. Please find enclosed our point-by-point responses according to your suggestion below:
Abstract
The abstract is short and could benefit from a concluding statement.
>> We have extended the abstract and have added a concluding remark.
Introduction
Overall, the introduction is very short and needs to be extended. Actually, almost 1/3 is description of method and should be moved to the section Method (p.2, line 52-62).
The text needs to be revised and I suggest language revision by native English speaker.
- 1, line 39: face protection; do you mean personal protective equipment or only face masks or shields? Please clarify
- 1, line 40-41: the phrase ‘…are often not affected severely…’. Do you mean seldom?
>> We have extended the Introduction but we do not agree that parts of the methods were described in the introduction. In the introduction we described the aim of our study. In the methods we described how we managed it. We corrected the term “face protection”. Thank you for the hint! We did not want to discuss if young and healthy people did experience severe symptoms rarely, but indeed, that they often did not experience severe symptoms: mild or even asymptomatic cases occur often in this group. This is good for them but makes them a danger for more vulnerable persons in their vicinity.
Method
Please clarify whether the four regencies mentioned as study locations are urban or rural areas.
Overall, this section needs to be restructured, so that information about the specific questionnaire is presented in one place which should be clearly marked with a sub-heading. As it stands, information about the questionnaire is found in the introduction (se the above comment) and in the section on study location and design.
- 2, line 76 refers to ‘in line with the attitude part of the questionnaire…’ which seems to imply, that the reader already knows about the different parts of the questionnaire. However, this information has not been provided. Please be aware of structure and clarity which is needed to follow the study transparency.
Related to most of the questionnaire information about the scores needs to be explained. Otherwise, the scores provided in Table 1 are quite difficult to follow. This includes the Indonesian baseline health standards score, please provide additional information about what low and high scores would indicate.
- 2, line 85… please rephrase. The questionnaire is not ‘asking questions.
>> Indeed, the four regencies include both urban and rural parts. We have added this information to the text. In the introduction, we only explain that this is a survey-based study. The details of the questionnaire are presented in the methods section. We have adapted the title of the sub-chapter. We have re-shuffled the order of referring to the other part of the questionnaire and have also rephrased the words in line 85 (original version). The scoring is explained in the annexes and we refer to the annexes in the methods section where appropriate.
Analysis
Please clarify what is meant by ‘factors that at least displayed a tendency of influencing knowledge’. Please clarify the choice to include factors influencing knowledge (p= 0,2) and which p-value levels were considered to indicate a statistically significant result?
>> Choosing a p-value of 0.2 as cut-off for inclusion in multiple regression was really an arbitrary, although reasonable ad-hoc decision. Indeed, most variables not included had a p-value in univariate analysis of 0.5 or higher. As usual, statistical significance was considered below a two-sided p-value of 0.05.
Results
As it stands, the results are structured under headings related to the statistical analysis performed. It would be more informative to present the results under headings that indicate what was actually investigated (e.g. Factors influencing COVID-19 knowledge).
The first part of the results seems to be focused more on describing the statistical analyses performed rather than focusing on what was actually found. Perhaps because the results were not statistically significant?
It would be more relevant to present actual findings and whether these were regarded as significant or not, and what this would mean in terms of clinical relevance – does it mean anything in real life?
The content of the section ‘3.3 Multinomial logistic regression’ provides much more relevant information and is much easier to follow.
>> We corrected the headings of the sub-sections. Yes, the methods begin with the descriptive statistics. This is not an uncommon way to do it. We now also provide percentages where appropriate. Analytical statistics and the significance of findings are presented in the next tables (and figures).
Discussion
This section could benefit from language revision. Please be mindful of the language used e.g. phrasing such as ‘better knowledgeable people’.
The discussion refers to one ‘most important finding of the study’, which is ‘that better knowledge is important for responsible health behavior that is both beneficial for oneself and for the community. However, I believe it is very important to remember, that answering a questionnaire at best indicates intensions to act accordingly to what is answered. It provides no data on what the participants actually did or would do in real life. Hence you should be careful to draw any conclusions related to actions and only refer to this finding as ‘participants intentions to act’. Hence, I do not think the discussion related to health literacy is relevant. I suggest focusing discussion on issues that can help us understand why participants provided the answers they did – what might have influenced their answers – and what can actually be drawn from this study?
>> We still hold that this finding is the most relevant finding of our study. But we acknowledge (also in the discussion) not that only having stated opinions from responses to a questionnaire, but no observed behavior, is a limitation of our study. This is a limitation, but it cannot be helped.
Conclusion
This needs to be revised based on a revision of the discussion.
>> As we did not change the main message of our discussion, we feel it is not necessary to alter the conclusion substantially.
Reviewer 3 Report
Review ijerph-2175350
In their manuscript "Determinants of COVID-19 Knowledge, Perception and attitudes in Indonesia: A Cross-Sectional Survey" Asyary et al. report on responsible behaviour against COVID-19 transmission in correlation with measured socioeconomic factors.
The article is well written and clearly presented. Minor revisions are required, after which the article should be published.
Formal
- Within parentheses square brackets should be used. Please edit this throughout the manuscript.
- Line 72 (Methods, sub header 2.1): There is an isolated closing bracket at the end of this sentence.
Minor
- Lines 60 following (Introduction): Was the information campaign originally in the public domain (e.g., a website)? If so, please provide a citation or link.
- Lines 72/73 (Methods, sub header 2.1): Please elaborate on methods of advertisements (e.g., word of mouth, flyers) and type of school to give inside toward potential selection biases.
- Table 1: Items of the male and female subgroups should not only list means and standard deviation, but also representation as percentages. Please include them.
- Figure 1: Confidence intervals can only be guessed from the current scale. I recommend using a more detailed scale.
- Line 175 following (Discussion): On which bases was prevalence in social media at the time established? Especially was a formal approach used? If so, this should be included in the Methods section, or a citation should be provided. Anyhow this information (be it rule of thumb or not) would be better suited in the Introduction or Methods.
Author Response
Reviewer 3
Review ijerph-2175350
In their manuscript "Determinants of COVID-19 Knowledge, Perception and attitudes in Indonesia: A Cross-Sectional Survey" Asyary et al. report on responsible behaviour against COVID-19 transmission in correlation with measured socioeconomic factors.
The article is well written and clearly presented. Minor revisions are required, after which the article should be published.
>> Thank you for this positive feed-back!
Formal
- Within parentheses square brackets should be used. Please edit this throughout the manuscript.
- Line 72 (Methods, sub header 2.1): There is an isolated closing bracket at the end of this sentence.
>> We tacked the problem of “parentheses in parentheses”. Thank you for pointing this out! We also corrected the error of the single closing bracket.
Minor
- Lines 60 following (Introduction): Was the information campaign originally in the public domain (e.g., a website)? If so, please provide a citation or link.
- Lines 72/73 (Methods, sub header 2.1): Please elaborate on methods of advertisements (e.g., word of mouth, flyers) and type of school to give inside toward potential selection biases.
- Table 1: Items of the male and female subgroups should not only list means and standard deviation, but also representation as percentages. Please include them.
- Figure 1: Confidence intervals can only be guessed from the current scale. I recommend using a more detailed scale.
- Line 175 following (Discussion): On which bases was prevalence in social media at the time established? Especially was a formal approach used? If so, this should be included in the Methods section, or a citation should be provided. Anyhow this information (be it rule of thumb or not) would be better suited in the Introduction or Methods.
>> We explained the information campaign and the methods of advertisement in more detail. We included percentages were appropriate.
Exact point estimates or confidence intervals are always difficult to read from a figure. A figure better serves to illustrate the main trends at a glance. We added exact (up to 2 decimal points) values or point estimates and confidence intervals in the text now.
Prevalence of social media use is based on other studies which we cite in the text.
Round 2
Reviewer 2 Report
To the authors,
Thank you for revision parts of the manuscript. However, as it stands - and based on your choices not to clarify issues pointed out to you as insufficiently desclosed in the previous versions e.g. related to method, presentation of results and the discussion, I recommend major revisions based on these issues once again. Pleace keep in mind, that these issues was raised based on concerns about whether an international audience would be able to follow your study and recieve it as trustwurthy and sound research - and not just comments made to pleace my personal interest. This is what peer-review is set in place for.
The arguments made for not addressing several of the issues raised does not speak in favor of making this manuscrpit a sound presentation of your work, e.g. the manuscript needs to be able to stand without the annekses (which should be used for additional nice-to-have information only). In addition to your responce to this part of the review, what was asked for was a presentation of clinical relevance - this unclarity is important to address - and could be improved if the questionnaire was described properly/sufficiently in the methods section.
These comments refer to just one example of where I believe the issues raised in the first round of review has not been adressed sufiiciently.
This is an interessting study, there is a need to work on the manuscript to strengthen transparency and rigor.
Author Response
I am a bit lost regarding the last proposals of the reviewer. The first paragraph only states that the review is an important and relevant statement that should be taken in account. It seems I offended the reviewer. For that I am deeply sorry.
The second paragraph raises 2 issues. The third paragraph states that these are only examples and the fourth states that overall, the study is interesting.
Therefore, I have to go with the 2 issues mentioned in paragraph 2:
(a) do not describe methods in the appendix but in the methods section!
(b) present the clinical relevance
(Indeed, the text is rather unclear, but these are the 2 points I make out!)
In the first round we had 3 reviewers. While 2 only suggested minor changes which we have accommodated, this 3rd reviewer (reviewer #2) also made more complex suggestions. We have fulfilled most of his (or her) requests. We did not: (i) shift parts of the introduction into the methods section. I still do not see which part of the introduction would describe methods. (ii) we did not move the tables from the appendices into the methods section. We could do it. But I believe this is up to the editor's decision. I do find that the details like the wording of the questions (a translation only anyway) would not fit into the main text. I do agree that we should make the weighting of the scores more clear and provide another reference for the choice of weights in the methods section. But if we should include the list of all items from the questionnaire with the scores and weights for every single item in the methods or if keeping these details in the appendices, is up to the editor to decide! I called upon editorial guidance but received no answer! (iii) we did not change what we perceived as the "most important finding of our study". Our study found that some personal characteristics predict better knowledge about COVID-19. But the impact of these characteristics (gender, housing conditions, number of diseases) on knowledge is not so strong and maybe not so important. But we found that knowledge, independent of these personal characteristics, also makes respondents at least state that they would do the right thing. I do know that knowing the right thing and saying the right thing is not the same thing as doing the right thing. We acknowledged that as a shortcoming of our study. But I hold that a person who knows what would be right is also more likely to do what is right. Therefore, I insist that our study indicates that better knowledge (about COVID-19 causes and symptoms) will also help people making the right choices when they are infected themselves. Maybe this is also not a very surprising finding. But from the public health perspective - and given the many fake news and misinformation around the pandemic, it is important to point out that proper information about the disease is relevant. We do discuss this under the theme of "health literacy". Our references show that here we follow the train of thought of other researchers. I do understand that health literacy is more than just "knowledge about COVID-19 causes and symptoms". But I find it is a good idea to embed our finding in this broader light.
If I understand the reviewer correctly, (and - as I told you - I find his 2nd report very difficult to understand!), then he criticises our response to items (ii) and (iii). As I said, I did extend the methods description regarding the scoring system and the weighting of the items for household characteristics (and assets). I call for an editorial decision as to whether I should move the tables from the appendices to the methods section (this, indeed, is an editorial decision!), and also I ask for editorial advice as to whether it is really necessary to write a completely different discussion. If the editor believes that a different discussion is required, the editor should also tell me what we should discuss if not the importance of knowledge for a health-promoting behaviour? While the reviewer clearly stated that our approach is not wanted, he (she) failed to suggest any clear alternative approach. I have no idea at all what else but their knowledge made participants answer as they did. In the first report reviewer 2 wrote: "I suggest focusing discussion on issues that can help us understand why participants provided the answers they did – what might have influenced their answers – and what can actually be drawn from this study?" As described, we have explored all other factors for which we had data and knowledge was the strongest and clearest determinant.
Round 3
Reviewer 2 Report
..
Author Response
since the reviewer has not made any detailed suggestions, there is nothing more for me to add.